# Next-Generation Sequencing for Venomics: Application of Multi-Enzymatic Limited Digestion for Inventorying the Snake Venom Arsenal

**DOI:** 10.3390/toxins15060357

**Published:** 2023-05-25

**Authors:** Fernanda Gobbi Amorim, Damien Redureau, Thomas Crasset, Lou Freuville, Dominique Baiwir, Gabriel Mazzucchelli, Stefanie K. Menzies, Nicholas R. Casewell, Loïc Quinton

**Affiliations:** 1Laboratory of Mass Spectrometry, MolSys Research Unit, University of Liège, 4000 Liège, Belgium; dredureau@uliege.be (D.R.); thomas.crasset@student.uliege.be (T.C.); lfreuville@uliege.be (L.F.); gabriel.mazzucchelli@uliege.be (G.M.); 2GIGA Proteomics Facility, University of Liège, 4000 Liège, Belgium; d.baiwir@uliege.be; 3Centre for Snakebite Research and Interventions, Liverpool School of Tropical Medicine, Pembroke Place, Liverpool L3 5QA, UK; stefanie.menzies@lstmed.ac.uk (S.K.M.); nicholas.casewell@lstmed.ac.uk (N.R.C.)

**Keywords:** venomics, proteomic, mass spectrometry, snake venom, toxin, multi-enzymatic digestion

## Abstract

To improve the characterization of snake venom protein profiles, we report the application of a new generation of proteomic methodology to deeply characterize complex protein mixtures. The new approach, combining a synergic multi-enzymatic and a time-limited digestion (MELD), is a versatile and straightforward protocol previously developed by our group. The higher number of overlapping peptides generated during MELD increases the quality of downstream peptide sequencing and of protein identification. In this context, this work aims at applying the MELD strategy to a venomics purpose for the first time, and especially for the characterization of snake venoms. We used four venoms as the test models for this proof of concept: two Elapidae (*Dendroaspis polylepis* and *Naja naja*) and two Viperidae (*Bitis arietans* and *Echis ocellatus*). Each venom was reduced and alkylated before being submitted to two different protocols: the classical bottom-up proteomics strategy including a digestion step with trypsin only, or MELD, which combines the activities of trypsin, Glu-C and chymotrypsin with a limited digestion approach. The resulting samples were then injected on an M-Class chromatographic system, and hyphenated to a Q-Exactive Mass Spectrometer. Toxins and protein identification were performed by Peaks Studio X+. The results show that MELD considerably improves the number of sequenced (*de novo*) peptides and identified peptides from protein databases, leading to the unambiguous identification of a greater number of toxins and proteins. For each venom, MELD was successful, not only in terms of the identification of the major toxins (increasing of sequence coverage), but also concerning the less abundant cellular components (identification of new groups of proteins). In light of these results, MELD represents a credible methodology to be applied as the next generation of proteomics approaches dedicated to venomic analysis. It may open new perspectives for the sequencing and inventorying of the venom arsenal and should expand global knowledge about venom composition.

## 1. Introduction

The recent and frequent advances in omics methodologies over the past two decades have greatly expanded our knowledge about biological systems. The integration of the constantly increasing amount of information stimulated by the fast advances of molecular biology, proteomics and bioinformatics has led to in-depth characterization of biological systems (tissue biopsies, cell cultures, biological fluids) at different levels (DNA, RNA, proteins, lipids) in a cost- and time-efficient manner [1].

One of the most applied omics methodologies is proteomics, which is usually supported by protein databases available either from public repositories or, more relevantly for venomics, from sample-relevant venom gland transcriptomes. The term proteomics describes a bioanalytical discipline that aims to determine the protein content of various samples, including their intrinsic modifications, conformations, interactions, abundances, activities, and cellular localizations [2,3,4]. The major analytical platform to achieve proteomic study relies on a coupling between a liquid chromatography chain and a mass spectrometer, both of which impact the quality of the data acquisition. Improvements in sample preparation protocols and the quality of the protein database used additionally increase the robustness of the method and the confidence of the results obtained [5].

Two different main ways to conduct proteomics experiments are classically explored: (1) top-down, which targets intact protein ions; and (2) bottom-up, which analyze peptides produced from proteolytic digestions [4,6]. Traditional bottom-up proteomics workflows involve the following steps: extraction of the protein mixture from the biological medium, reduction and alkylation of the potential disulfide bonds, proteolytic cleavage usually using trypsin, followed by a LC-MS/MS of the resulting peptides, then by a database search for protein identification [3]. The recent improvements of MS instrumentation (resolving power, mass accuracy, ion activation, detection, etc.) facilitate high quality data generation. However, major limitations often occur, such as the loss of the smallest and the most hydrophilic peptides during the chromatographic step, the sequence-dependent efficiencies of digestion, non-comprehensive datasets due to the oversampling of the most abundant peptides, or again the lack of uniform cleavage site distributions within the protein sequences [4,7].

Indeed, almost all the digestion protocols of relevance to the bottom-up proteomic approaches described above involve the application of trypsin to reach a complete digestion, which generates up to 3 kDa peptides [8]. In 2019, Morsa et al. [4] described a new generation of sequencing for proteomics, named “Multi-Enzymatic Limited Digestion” (MELD), which presents a single 2 h-proteolytic step based on the synergic action of a diluted mix of enzymes (Trypsin, Chymotrypsin and Glu-C). The low enzymatic concentration and the restricted reaction time induce a limitation in the proteolytic processes and a high content of missed cleavages, resulting in the generation of peptide products of diverse lengths with overlapping sequences. This improves the sequence coverage and the confidence of protein identification, as well as the efficiency of database-free *de novo* sequencing [4].

The application of new methodologies for proteomics is especially interesting to the studies of animal venoms. The complex mixtures of diverse, selective and potent natural products found in animal venoms has garnered significant interest as a source of potential therapeutic leads [9]. Further, understanding the variable compositions of animal venoms is extremely important for defining the evolutionary processes driving venom variation, understanding the symptoms observed in human victims of animal envenoming and for informing the design of more specific antivenoms to combat these pathological effects [10]. Considering that envenoming, particularly snakebite envenoming, represents a global public health problem defined by the WHO as a priority Neglected Tropical Disease [11], studies that seek to characterize and inventory in-depth the toxins found in animal venoms are extremely important. Indeed, the breakthrough experienced by venom research in the last decade is largely due to the development and application of omics technology to the qualitative and quantitative profiling of venom and venom gland composition. Thus, a new term was coined to combine this comprehensive workflow applied to venoms, named venomics [12]. In a brief search on PubMed, over the past 10 years there were 1135 proteomic studies related to venoms compared with 831 studies focused on venom gland transcriptomes, which shows that proteomics is the omic technique most readily applied in venomics studies.

Traditionally, the protein content of venoms is primarily composed by peptides from 5-kDa to >100 kDa-proteins and there are particularities for the sequences of these toxins for some species. In some Elapidae snake venoms, the main group of toxins, namely Kunitz-type toxins, are rich in basic amino acid residues [13]. Knowing that trypsin cleaves specially in the C-terminus extremity of K and R residues, strong oversampling appears, which creates a decrease in sequence coverage of the toxins, leading to a bias in the data analysis. The application of MELD to venomics should help to overcome such challenges by increasing the coverage of the sequencing of toxins found in these complex mixtures. Therefore, herein we explore for the first time the application of a MELD protocol for the characterization of snake venoms, using representatives of two different snake families (Elapidae and Viperidae) as our model, and we demonstrate the feasibility and superiority of this approach over traditional bottom-up proteomics for venomics purposes.

## 2. Results and Discussion

### 2.1. Data Acquisition

To study the impact of the digestion protocol on the sequence coverage of snake toxins, the two following approaches were applied: mono-enzymatic digestion (classical bottom-up) and Multi-Enzymatic Limited Digestion (MELD). For a proof of concept, we used the venom from two Elapidae (*Dendroaspis polylepis*, DpV and *Naja naja*, NnV) and two Viperidae (*Bitis arietans*, BaV and *Echis ocellatus*, EoV). First, a peptide mass fingerprint was generated in the mass range of 400–6000 *m*/*z*, with MALDI-TOF/MS (HCCA matrix). We observed that differences between the two protocols did not substantially impact on the resulting number of ions captured in this mass range, although some were found to have increased resolution and intensity with MELD compared to the traditional trypsin-based bottom-up protocol (Appendix A).

The application of high throughput techniques has emerged to increase the comprehension of the peptide sequences from an organism in a faster and deeper way. In shotgun proteomics, the injection of a crude venom digest into (nano)LC-MS/MS for analysis proves to be less laborious, faster, and still provides a global view of the toxin composition in a venom of interest [14]. The combination with a reliable software for data analysis (Peaks X+ in this study) allows result matching, i.e., the exact masses combined to the MS/MS spectra, with the protein database, and to predict toxins sequences based on *de novo* sequencing when not present in the database. It also permits researchers to propose putative post-translation modifications not easily achievable when performing top-down proteomics [14,15]. There are two main ways to perform the peptide sequencing: database search and *de novo* sequencing. Database search queries a sequence database for the peptide sequence which explains best the peaks observed in the MS/MS spectrum, whereas *de novo* sequencing proposes the most probable peptide sequence explaining the MS/MS spectrum, which is only based on the exact mass of the precursor and of the fragment ions [15].

Overall, an average of 506 (±198) peptides were reported in the trypsin protocol against 1130 (±705) peptides for MELD, this technique being superior for the unique peptide identification around 2.2-time (*p* = 0.0074). The MELD protocol resulted in significant increases in the number of peptides identified from the database for three of the four venoms, with only a modest, non-significant increase observed for the DpV (Figure 1A). The fold increases observed were 1.2-fold for DpV, 2.6-fold for NnV, 1.7-fold for BaV and 2.7-fold for EoV. The increases observed are coherent with the fact that MELD, by exploiting the properties of three different enzymes, generates more peptides than the classical bottom-up strategy. This larger library of peptides logically offers more possibilities of matching peptides from the reference database. It is important to note that the figure displays the numbers of unique peptides only, which represents the supporting peptides that are only linked to one protein group. These peptides are most useful for protein identification based on database search.

The *de novo* peptides (Figure 1B) are represented by all the confident *de novo* sequence tags that remain unidentified in the database. This can be caused, for example, by the absence of a detailed protein database for the species of interest (for example, when no transcriptomic data is accessible), by the existence of errors in the database of reference or by the presence of unexpected post-translational modifications. To be considered a *de novo* peptide, the score must be equal to or better than the specified threshold applied in the software, as described in the methodology section. For the trypsin protocol, an average of 5274 (±1654) *de novo* peptides were found in total, while for MELD, 6090 (±1378) *de novo* peptides were reported (*p* = 0.2029). This represents a slight increase, but it is much more significant when we evaluated it separately by venom.

Figure 1B shows that the number of remaining *de novo* peptides barely increased for three of the four species, though only two of these resulted in statistically significant increases (NnV, EoV). MELD improved the *de novo* identification by 1.2-fold for DpV, 1.1-fold for NnV and 1.5-fold for EoV, while the trypsin-based bottom-up approach resulted in a 1.1-fold increase for BaV.

Dau et al. (2020) [16] analyzed the characteristics of five protease alternatives (AspN, Glu-C, chymotrypsin, elastase and proteinase K) to trypsin for protein identification and sequence coverage. The authors found a lower number of identified MS/MS spectra for these enzymes, used individually, compared to trypsin. Contrary to the current study, Dau et al. applied the enzymes separately in one single reaction. However, protein and peptide identifications are enhanced when combining any of the tested proteases with trypsin in a sequential reaction [16]. This shows that the combination of different enzymes in the same digestion can improve the yield of MS/MS data acquisition and database matches, which will impact the resulting sequence coverage.

### 2.2. Database Search

The analysis of proteomics results using the protein database gives similar outcomes concerning the number of identified sequences, namely the total number of unique sequences, excluding I\L and PTM duplications (Figure 2). An average of 488 (±190) sequences were identified in total with the classical bottom-up approach. When MELD is used, these numbers are significantly increased overall (an average of 1092 ± 672.8 representing a 2.2-time fold increase; *p* = 0.0067), dictated by significant increases observed for NnV (2.6-fold), BaV (1.8-fold) and EoV (2.6-fold) venoms, though only slight increases were observed for DpV (1.1-fold) (Figure 2), in line with our peptide recovery findings (Figure 1). The observation that DpV is not beneficially affected by MELD could possibly be explained by its venom composition; visualization on SDS-PAGE (Figure 2A) demonstrates that this venom is particularly dominated by lower molecular mass proteins, mostly falling in the mass range of 6–14 kDa. Therefore, it seems likely that there are fewer sites to be cleaved by the enzymes employed for digestion. Although NnV belongs to the same Elapidae family as DpV, it has been found to benefit from the technique. This is likely due to the presence of high-molecular-weight molecules (ranging from 28–98 kDa) in the venom of NnV, as illustrated in Figure 2A.

The sequences constitute the basis for the identification of proteins (Figure 2B), and the software proposes the following classification: protein groups (Figure 2C) and top proteins (Figure 2D). Protein groups are the total number of groups of proteins with significant peptides, which are the peptides that pass the selected filters. The significance score is calculated as the −10logP of the significance testing *p*-value. In this way, MELD had better results for NnV, BaV and EoV compared to trypsin, varying from a 1.2 to 1.5-fold increase. “Top proteins” designates the proteins supported by the highest number of unique peptides in the protein groups [17]. Consequently, a protein validated by a peptide not supporting the top proteins will be added to a new group of proteins. For those parameters, we observed a tendency of improvement (from 1.1 to 1.3-fold) with the MELD protocol, as evidenced by the data for NnV, BaV and EoV venoms (Figure 2D) compared to trypsin digestion protocol.

Figure 3 shows the classification of the peptide and protein sequences whether detected in trypsin-only, in MELD, or in both approaches (in common). As expected from the previous results (Figure 2), DpV is the less affected by MELD as the number of identified proteins is lower compared to the trypsin-based approach (Figure 3A). On the contrary, for NnV (Figure 3B), BaV (Figure 3C) and EoV (Figure 3D), MELD strongly increased the number of peptides and proteins identified from the protein database. A complete comparative list of peptides and proteins for each snake is available in the Appendix A.

### 2.3. Venom Composition

To perform the venom component classification, we considered a similar grouping proposed by Damm et al. (2021), combined by Boldrini-França et al. (2016), for major and minor toxins [18,19]. Figure 3 shows that trypsin and MELD are complementary to identify the proteins from the database, for all the venoms. In a deeper analysis, Figure 4 compares these approaches for the identification of the major toxins, but also for minor toxin proteins and proteins of lower abundances, such as cellular components and additional proteins, for which no clear biological role has been defined to date. The main observation is that, again, both methodologies give globally similar results, with most toxins found in each venom, demonstrating their applicability for providing in-depth coverage of the venom toxin arsenal. It is important to remark that DpV, for which the use of MELD did not enhance the number of identified proteins compared to trypsin-only, an improvement of the proportion of major toxins identified was observed (30.4% to 47.6%). BaV and EoV showed the best results, with more than 80% of sequences for major toxins identified, compared to 73.4 and 51.5%, respectively, for the trypsin protocol. Followed by NnV, MELD identified 77.2% of major toxins against 53.4% for the trypsin methodology.

Table 1 displays the relative distribution of the 26 components analyzed among the four venoms, with the trypsin and MELD protocols. From those, 20 show to be much more characterized with the MELD protocol, especially the 3-finger toxins, venom kunitz-type, phospholipase A2, snake venom serine proteases (SVSP) and snake venom metalloproteinases (SVMP), which collectively represent the major toxin families found in Elapidae and Viperidae snake venoms. In addition, MELD decreased the proportional representation of cellular components and protein family not assigned in all venoms (Figure 4). Moreover, some components, such as bradykinin-potentiating peptides, cathepsins, vascular endothelial growth factors and phospholipases B were detected in some venoms only when the MELD protocol was applied, such as for NnV and BaV. These data demonstrate that the MELD methodology improves the diversity of identified toxins and, consequently, the quality of venom arsenal composition.

### 2.4. Sequence Coverage of Major Toxins

To evaluate how the trypsin and MELD protocols impact the sequencing of toxins, the most abundant toxins from each venom were selected and sequencing parameters were compared (Table 2). These toxins are the 3-finger toxin alpha-elapitoxin-Dpp2d (Uniprot accession C0HJD7) for DpV, the 3-finger toxin Cytotoxin 10 (Uniprot accession P86541) for NnV, the C-type lectin Bitiscetin-3 subunit alpha (Uniprot accession A0A5A4WNG2) for BaV and the SVMP Group III snake venom metalloproteinase (Uniprot accession E9KJY6) for EoV. MELD showed an improvement for almost all of the six parameters studied (peptide coverage, number of peptides, number of *de novo* tags, -10logP, number of spectra and number of unique peptides).

For sequence coverage, MELD obtained better results for all toxins, except for cytotoxin 10, where the same coverage (100%) was observed for both protocols. The number of peptides and unique peptides relating to the four toxins was also increased with MELD. For NnV and EoV, MELD improved all the parameters, whereas for DpV only the number of spectra were not increased by this protocol. This constitutes a good example of how the oversampling of trypsin digestion can affect the sequence coverage. Although Dpp2d had a higher number of spectra with trypsin digestion, the sequence coverage of this toxin is better with MELD (82% with trypsin against 89% with MELD). The same behavior is observed with Bistiscetin-3 from BaV (67% vs. 75%).

As mentioned earlier, several proteomic strategies have been historically applied to characterize venom composition, with bottom-up approaches dominating until recently. A recent study of Tasoulis et al., 2022 compared different methodologies used in 67 studies investigating snake venom proteomics [20]. The authors showed that the most common workflow used in around half of the studies to characterize the venom composition was based on a previous fractionation by RP-HPLC, followed by 1D SDS-PAGE, in-gel trypsin digestion and mass spectrometry. They reinforce how a preliminary decomplexation of crude venom by chromatography is important in order to obtain an in-depth overview and an increase of the protein identification quality.

Indeed, the combination of different methodological strategies for proteomics and integrating venom gland transcriptomics and public databases will maximize proteome coverage [20,21,22]. However, it is important to highlight that the majority of venomic studies apply digestion with trypsin, which is considered the gold standard in proteomics. In this context, modifications in the protocol applying the combination of multiple enzymes offers several advantages over the traditional use of trypsin in sequencing toxins. Not only is it easier and faster, but it also proves to be the most cost-effective and highly efficient approach. By implementing this alternative enzymatic method, significant improvements can be achieved in proteomic analysis, surpassing the limitations of analytical equipment utilized in the past two decades. [4]. The need for scaling up the analysis with a high-throughput workflow is also something to be considered in order to obtain a huge amount of data in a less laborious manner and without losing the quality of the obtained sequences.

The monoenzymatic reactions present a sequence-dependent efficiency and also require a uniform cleavage site within the sequence (C-terminal side of lysine and arginine residues for trypsin). Nevertheless, the application of one enzyme can lead to an oversampling of the most abundant peptides [4]. The application of different enzymes can increase the number of proteolytic sites leading to a higher number of peptides, which can directly impact the sequence coverage and therefore improve the characterization of PTMs, as we observed in this study (Table 2). In this sense, MELD presents several advantages, not only because of the reduced reaction time (2 h) compared to trypsin digestion (usually done overnight), but also because of the specificity of the cleavage sites. MELD exploits three enzymes: (1) Chymotrypsin (EC 3.4. 21.1), which is a 26 kDa serine carboxypeptidase that preferentially cleaves the amide bond (the P1 position) of an aromatic amino acid residue such as tyrosine, tryptophan and phenylalanine, and to a lesser extent, leucine and methionine [23,24]; (2) Glu-C, which is a serine protease which cleaves at the C-terminus of glutamic and aspartic residues depending of the buffer pH. In ammonium acetate pH 4.0 or ammonium bicarbonate pH 7.8, the enzyme preferentially cleaves glutamyl bonds. However, in phosphate buffer pH 7.8, Glu-C is reported to cleave at either site [25,26]; and (3) Trypsin, which is also a serine protease that specifically cleaves peptide bonds at the C-terminal side of lysine and arginine residues, except for -Arg-Pro- and -Lys-Pro- bonds, which are normally resistant to proteolysis [27].

The vast application of trypsin in proteomics stems at least its high activity and selectivity at physiological pH and also partly from the resulting digested peptides presenting a positive charge at the peptide C-terminus, which is advantageous for MS analysis (better ionization and easier MS/MS spectra interpretation). Another advantage is that trypsin generates short peptides with a basic Arg or Lys at the C terminus, which is useful for chromatographic separation and search algorithm-based identification and quantification methods. However, due to the short length of the digested peptides, they may not be identified by MS, meaning that only a restricted segment of the proteome is covered [23,28]. Therefore, the addition of other enzymes may increase proteome coverage. MELD applies chymotrypsin and Glu-C combined with trypsin in the same reaction. Chymotrypsin has been used in combination with trypsin or other proteases to provide additional sequence information for protein regions that are not favorable to trypsin [23]. Glu-C cleaves mainly at the C-terminal side of Glu and its specificity depends on the buffer composition and pH. An advantage of Glu-C is that the digestion is not modified by glycation close to Glu and Asp, increasing its application [23]. Thus, the combination of trypsin with Chymotrypsin and Glu-C in a single reaction can considerably improve the results for proteomics, especially for venomics in which glycosylations are a common post-translation modification found in toxins.

Related to venomics, researchers have demonstrated that using a single method (in gel digestion, bottom-up/top-down proteomics, pre-fractionation, etc.) to estimate snake venom composition can result in incomplete proteomic coverage and that combining methods may be preferred [29,30]. Some recent studies have compared different methodological steps and demonstrated a concerning degree of variation in the results [29,31]. However, all the studies reviewed by Tasoulis et al. (2022) presented something in common: the utilization of only trypsin for protein digestion. Therefore, as far as we know, this work constitutes the first study that exploits an enzyme combination to reveal venom proteomes. Most venomics studies have focused on the abundance of various toxins and toxin families, and diversity is often not reported or investigated. According to our results, the application of mono-enzymatic digestion can directly impact this observation made by Tasoulis et al. (2022) [20] and even create a non-desirable bias.

The improvements in the protocols for venomics have to go beyond the analytical methods, and MELD can increase the toxin diversity and provide significant insights into the biology and evolution of (snake) venoms. Diversity can be considered as either the number of different protein families expressed in a snake venom, or in terms of toxin diversity, being the number of toxins within these protein families [20]. Mono-enzymatic digestions in venomics can uncover the presence of multiple isoforms of the same toxin resulting from the evolutionary process, as overlapping fragments from the same cleavage site can be detected (i.e., -R and -K for trypsin-only digestions). It is possible that this phenomenon explains the observed results in DpV. This venom primarily consists of peptides, with a significant proportion containing basic amino acids. Similar observations have been made with other animal venoms. It is known that certain toxins, particularly those found in scorpions’ venoms, are abundant in basic amino acid residues. This abundance can result in the overlapping of small peptides when using trypsin alone for sample preparation, thereby leading to limited sequence coverage. [32]. This issue is theoretically overcome with the MELD protocol. Furthermore, venoms that are abundant in high molecular weight compounds may benefit from having more cleavage sites for the MELD protocol, leading to improved outcomes.

Finally, MELD offers the ability to set the software parameters for the database analyses to select “no enzymes” in the features. This helps to ensure coverage of other small peptides in the database search that may be lost when we set only “trypsin” as the digestion feature. Multi-enzymatic protocols in general were already applied to venomics of the spider *Loxosceles intermedia*, and the approach increased protein coverage [33]. Although the authors used different enzymes than us, and a venom that differs a lot from snake ones, we can observe how the changes in the sample preparation can lead to improved results for venom studies. We showed a proof of concept in which MELD not only improved the yield of data acquisition, but also presented more matches in the database search, coverage of sequences and in the identification of new venom components for snakes. This methodology may represent an innovative way for faster and deeper venomic studies.

Studies show the importance of the association of different analytical techniques for protein decomplexation methods to perform venomic studies, together with improvements in the database search using the transcriptome data and new software [18,20,34,35]. Indeed, they increase the sequence coverage, which helps to understand the venom composition. However, studies have yet to adopt the combination of multiple enzymes for sample preparation, such as MELD for snake venoms, as described here. Our findings demonstrated that incorporating a multiple enzyme protocol can positively impact venom characterization. Herein, we showed for the first time the proof of concept and advantages to apply the MELD protocol as a sample preparation to snake venoms for venomics studies.

## 3. Conclusions

We showed that MELD is likely to be of considerable value for venomics characterization of snake venoms. This methodology not only improved the yield of data acquisition, but also presented more matches in the database search, coverage of sequences and the identification of new venom components. This strategy can be applied to identify new groups of venom constituents and to increase the in-depth sequencing of toxins. Therefore, the MELD methodology may represent an innovation for venomic studies and may open new perspectives for sequencing and inventorying the snake venom arsenal in a faster and deeper way.

## 4. Materials and Methods

### 4.1. Venoms Selection

For these analyses, we aimed to evaluate how the digestion protocol would behave according to the sequence coverage for different venom profiles of snake venoms sourced from different families. To that end, we selected venoms from *Dendroaspis polylepis* (DpV) and *Naja naja* (NnV) as the Elapidae species, and for the Viperidae family, *Bitis arietans* (BaV) and *Echis ocellatus* (EoV) were chosen. DpV and EoV were provided by the Centre for Snakebite Research and Interventions, Liverpool School of Tropical Medicine (Pembroke Place, Liverpool, Liverpool, UK); BaV and NnV were purchased from the Alphabiotoxine Laboratory (Montroeul-au-bois, Frasnes, Belgium). The venoms were obtained in their lyophilized form and stored at −20 °C until use.

### 4.2. Polyacrylamide Gel Electrophoresis (SDS-PAGE)

Each venom sample (40 µg) was added to Laemmli buffer and heated at 100 °C for 3 min. Proteins were separated on a 4–12% NuPAGE MES gel (Thermo Fisher Scientific, Waltham, MA, USA) run at 200 V for 45 min. A molecular marker standard consisting of insulin beta-chain (3 kDa), aprotinine (6 kDa), lysozyme (14 kDa), red myoglobin (17 kDa), carbonic anhydrase (28 kDa), alcohol dehydrogenase (38 kDa), glutamic dehydrogenase (49 kDa), bovine serum albumin (62 kDa), phosphorylase (98 kDa) and myosin (188 kDa) was used. After the run, the gel was firstly dehydrated with 50% EtOH and phosphoric acid 3% (*v*/*v*) for 3 h, followed by a rehydration by 20 min bath of ultrapure water (MilliQ). Visualization of the proteins was performed overnight with Coomassie blue at 360 g/L, in an aqueous buffer with 34% MeOH (*v*/*v*), 3% phosphoric acid (*v*/*v*) and 17% ammonium sulphate (*v*/*v*). The gel was finally conserved at 5 °C in 5% of acetic acid (*v*/*v*) followed by a scanner record.

### 4.3. Digestion Protocols

In triplicate, 10 µg of each lyophilized venom was dissolved into 20 µL of 50 mM NH_4_HCO_3_ pH 7.8. The sample was reduced with 2 µL of 30 mM dithiothreitol (DTT) for 40 min at 56 °C under shaking at 650 rpm. The reduced samples were alkylated for 30 min at room temperature in the dark with 3 µL of 60 mM iodoacetamide. A second step with 2 µL of 60 mM DTT was performed for 10 min at room temperature in the dark for quenching the alkylation. After that, we proceeded with two different protocols for digestion: the classical with trypsin, named the mono-enzymatic protocol, and the synergic multi-enzymatic limited digestion (MELD).

#### 4.3.1. Mono-Enzymatic Protocol

For the classical protocol, the samples were submitted to two consecutive steps of digestion with trypsin: the first one at a ratio of 1/50 (trypsin:protein) with overnight incubation at 37 °C and shaking at 650 rpm. The next day, a second step was performed with a ratio of 1/100 (trypsin:protein) with acetonitrile added to a final concentration of 80% (*v*/*v*). The second step was incubated at 37 °C for 3 h. The reactions were quenched by acidification with 10% TFA at final concentration (*v*/*v*) to reach pH~3.0. The digested samples were dried on speed vacuum and resuspended in 20 µL of 0.1% TFA for desalting on ZipTip pipette tips with C18 resin. The elution was performed by 20 µL of water 0.1% TFA/ACN (50/50). The resulting eluates were evaporated under speed vacuum and reconstituted to 15 pmol/9 μL in H_2_O with 0.1% TFA for injection on the mass spectrometer.

#### 4.3.2. Synergic Multi-Enzymatic and Limited Digestion (MELD)

This protocol was developed by Morsa et al. [4], and applies three enzymes (trypsin, Glu-C, chymotrypsin) in the same reaction but in two parallel steps according to the ratio of the enzymes: high-ratio and low-ratio. The enzymes were prepared on ice immediately before use by mixing the pure 1 mg/mL solutions in a ratio of 1.00/1.00/1.55 (*v*/*v*) for trypsin/Glu-C/chymotrypsin. For the high-ratio MELD, the mixture was used *a priori*, and for the low-ratio MELD it was obtained by a 9-fold dilution of the former using 25 mM NH_4_HCO_3_ and 5 mM CaCl_2_. Two distinct digestions were performed simultaneously by, respectively, adding the same volume of the prepared mixtures to two protein fractions of 10 µg to obtain final protease-to-protein ratios of 1/85, 1/85, and 1/55 for the high-ratio digestion and 1/750, 1/750, and 1/500 for the low-ratio digestion by trypsin, Glu-C, and chymotrypsin, respectively. Each tube was incubated for 2 h at 37 °C under stirring at 650 rpm (Thermomixer Comfort, Eppendorf, Hamburg, Germany). The reactions were stopped using 10% TFA (*v*/*v*) at final concentration to reach pH~3.0. Equal amounts of both digests were subsequently pooled and the resulting mixture was evaporated under vacuum and reconstituted to 15 pmol/9 μL in H_2_O with 0.1% TFA for the injection on the mass spectrometer.

### 4.4. MALDI-TOF/MS

The peptide mass fingerprint of each venom after the digestions were obtained by Matrix-Assisted Laser Desorption/Ionization Time-Of-Flight Mass Spectrometry (MALDI-TOF/MS, RapiFleX, Bruker Daltonics, Bremen, Germany) using FlexControl 3.0 software for data acquisition. Of each venom, 0.175 µg was digested with MELD and trypsin were co-crystallized with α-cyano-4-hydroxycinnamic acid (HCCA) (40 mg/mL, prepared in 50/50 acetonitrile/H_2_O/0.1% formic acid *v*/*v*) using the dried-droplet method, directly on the MALDI plate. MALDI-TOF was calibrated with peptide I calibrant (Bruker Daltonics, Bremen, Germany) in reflective positive mode, for the molecular mass range from 400 to 6000 *m*/*z*. A total of 3000 mass spectra were recorded and analyzed using FlexAnalysis 3.4 software (Bruker Corporation, Billerica, MA, USA).

### 4.5. Liquid Chromatography—Mass Spectrometry Analysis

The LC-MS/MS analyses were performed on an Acquity M-Class UPLC (Waters) hyphenated to a Q-Exactive (Thermo Scientific, Waltham, MA, USA) for the digested samples from snake venoms in nano-electrospray positive ion mode. The trap column is a Symmetry C18 5 μm (180 μm × 20 mm) and the analytical column is an HSS T3 C18 1.8 μm (75 μm × 250 mm) (Waters, Corp., Milford, CT, USA). The samples were loaded at 20 μL/min on the trap column in 98% solvent A (0.1% formic acid in water) for 3 min and subsequently separated on the analytical column at a flow rate of 600 nL/min with the following linear gradient: initial conditions 98% A; 5 min 93% A; 60 min 70% A; 70 min 60% A; and 73 min 15% A maintained for 5 min, then the column was reconditioned in initial conditions. The solvent B is 0.1% formic acid in acetonitrile and the total run time is 100 min.

The mass spectrometer method is a TopN-MSMS method where N was set to 12, meaning that the spectrometer acquires one Full MS spectrum, selects the 12 most intense peaks in this spectrum (singly charged and unassigned charge precursors excluded) and makes a Full MS2 spectrum of each of these 12 compounds. For MS spectrum acquisition: mass range from 400 to 1750 *m*/*z*; resolution of 70,000 (at *m*/*z* 200); AGC target of 1 × 10^6^ or maximum injection time of 200 ms. The parameters for MS2 spectrum acquisition: isolation window of 2.0 *m*/*z*; Normalized Collision Energy (NCE) of 25; resolution of 17,500; AGC target of 1 × 10^5^ or maximum injection time of 50 ms. The main tune parameters for Q-Exactive: spray voltage of 2.3 kV, capillary temperature of 270 °C and S-Lens RF level of 50.0.

### 4.6. Data Analysis

Protein identification by automated *de novo* sequencing was performed using the software Peaks X+ version Studio 10.5 software [17] against the database created by the deposits related to “Snake + Venom” in the UniProt repository (74,759 sequences). Carbamidomethylation was set as fixed modification and oxidation (M) were set as variable modification, with maximum missed cleavages at 3. Parent mass and fragment mass error tolerances were set at 5 ppm and 0.015 Da, respectively. A false discovery rate (FDR) of 0.1% and unique peptide ≥1 with significant peptides were used to filter out inaccurate proteins for the PEAKS search algorithms and “*De novo* only” analysis with a −10lgP > 20 for the database match with high confidence. A protein score of 20 or higher is recommended and the score was −10 times the common logarithm of the *p*-value. The unique peptides are the high-confidence peptides that are unique to the group of proteins (not found in other protein groups). To achieve confident results, at least one unique peptide is needed for a protein group. The top proteins were used for the classification, these proteins are the components supported by the most unique peptides in the group. We grouped these components according to major toxins (with a known function in envenoming), minor toxins (potential bioactive molecule but unknown role in envenoming), cellular components and protein family not assigned (with unknown function on database). The venom component classes (major toxins and minor toxins) were also clustered according to their venom families. The percentage of the venom components in each digested venom was calculated as described by Abidin and colleagues [36], using the following formula:
$$[number\;of\;proteins\left(protein\;family\right)÷total\;proteins\;detected\;using\;LC-MS/MS]\times 100$$


### 4.7. Statistical Analysis

The experimental data are presented as mean ± SD. The data were analyzed by multiple Student t-tests using the software GraphPad Prism, version 8.0.2 for Windows (GraphPad Software, San Diego, CA, USA, 2019). The experiments were performed in triplicate, and all bars and written values are expressed as mean ± S.D. Values with *p* < 0.05 were considered statistically significant.

## Figures and Tables

**Figure 1 toxins-15-00357-f001:**
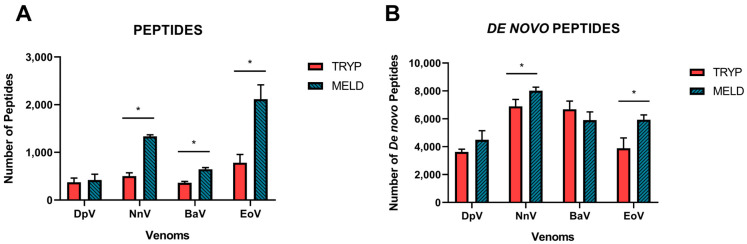
Data acquisition after applying the parameters in the software according to the snake venoms proteomics. (**A**) Number of peptides identified by the database search. (**B**) The *de novo* peptides predicted by the *de novo* sequence tags that remain unidentified in the database. The experiments were performed in triplicate, and all values are expressed as mean ± S.D. * *p* < 0.05 vs. Tryp group.

**Figure 2 toxins-15-00357-f002:**
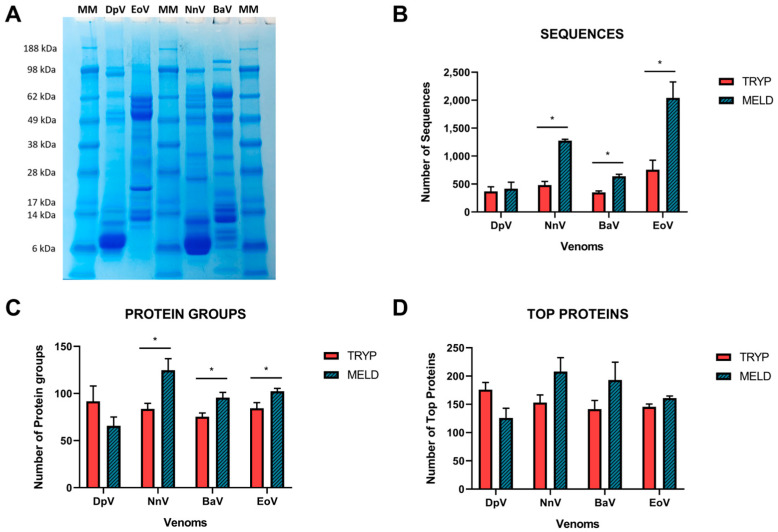
Overview of the database match according to the digestion protocol. (**A**) SDS-PAGE gel electrophoresis profile for each venom. (**B**) Number of sequences and matches of the database search according to the protein groups (**C**) and top proteins (**D**) among the analyzed venoms. The number of proteins is displayed in the Y axis for each venom. At least one unique peptide is needed for a protein group. The top proteins are the components supported by the most unique peptides in the group. The experiments were performed in triplicate, and all values are expressed as mean ± S.D. * *p* < 0.05 vs. Tryp group.

**Figure 3 toxins-15-00357-f003:**
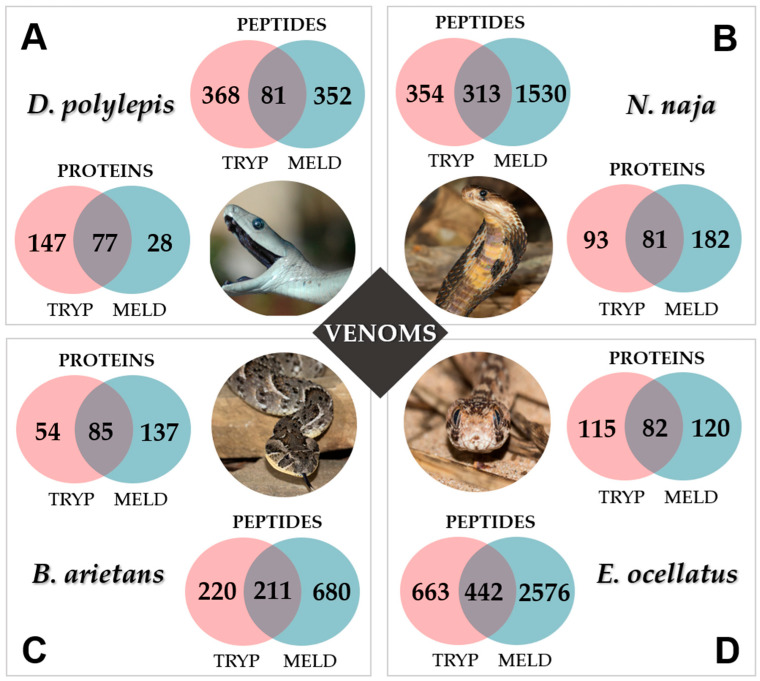
Venn diagram showing number of peptides and proteins identified and their shared molecules between each digestion protocol (MELD vs. Trypsin) for each venom: *D. polylepis* (**A**), *Naja naja* (**B**), *Bitis arietans* (**C**) and *Echis ocellatus* (**D**) venomsImages: Thea Litschka-Koen (https://eswatiniantivenom.org/; accessed on 3 April 2023) for Dp image and Ray Wilson (http://www.raywilsonbirdphotography.co.uk/index.html; accessed on 3 April 2023) for Nn, Ba and Eo snakes. A complete comparative list of peptides and proteins for each snake can be accessed in the Appendix A.

**Figure 4 toxins-15-00357-f004:**
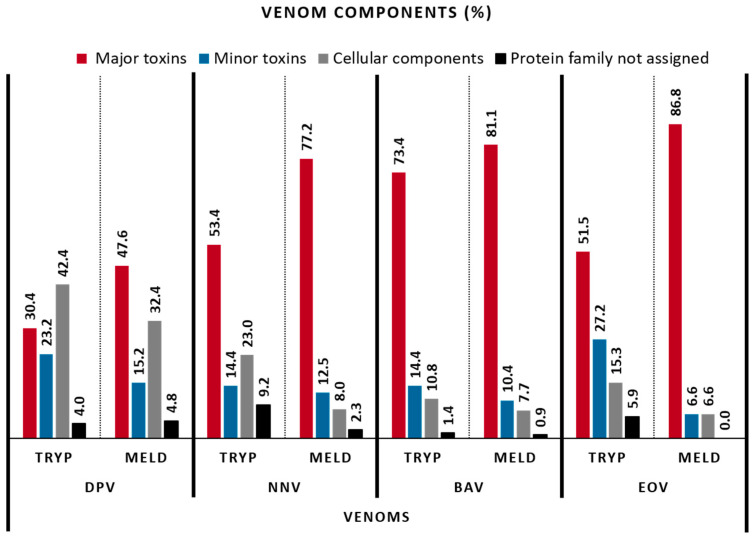
Relative venom composition among the venoms for the major toxins, minor toxins, cellular components and protein family not assigned.

**Table 1 toxins-15-00357-t001:** Venom components distribution among the venoms according to the protocol applied.

VENOM CLASSES (%)	VENOMS							
	DpV		NnV		BaV		EoV	
	TRYP	MELD	TRYP	MELD	TRYP	MELD	TRYP	MELD
Major toxins								
3-finger toxins	15.2	21.9	19.54	39.2	1.4	13.5	0	0
Snake venom metalloproteinases	6.3	13.3	10.34	11.8	27.3	18.9	23.8	48.2
Snake venom serine proteinases	6.3	1.0	5.75	5.7	12.9	14.9	9.4	14.2
Venom Kunitz-type family	1.3	6.7	3.45	2.3	0.7	1.4	0.5	1
Disintegrins	0.4	0.0	0.00	0.0	5.8	8.6	2	0.5
Natriuretic peptides	0.4	4.8	0.00	0.0	0.0	0.0	0	0
Phospholipases A2	0.4	0.0	5.75	14.8	8.6	4.5	1.5	2.5
Bradykinin-potentiating peptides	0.0	0.0	0.00	0.0	0.0	0.5	0	0
C-type lectin-like	0.0	0.0	4.60	0.4	9.4	14.0	9.9	17.8
Cysteine-rich venom proteins	0.0	0.0	2.87	2.3	5.0	3.6	3.5	1.5
L-amino-acid oxidases	0.0	0.0	1.15	0.8	2.2	1.4	1	1
Minor toxins								
Ohanin/vespryn family	5.4	0.0	4.02	0.8	0.0	0.0	14.4	0.0
Phosphodiesterases	4.9	1.0	1.15	1.1	2.2	1.8	2.0	1.5
Nerve growth factors	3.6	4.8	0.57	0.8	0.7	0.5	2.5	0.0
Hyaluronidases	2.7	4.8	0.00	0.0	0.7	0.0	0.0	0.0
Aminopeptidases	1.8	1.9	1.15	1.1	2.2	1.4	5.0	1.0
Venom endothelial growth factors	1.8	0.0	0.00	3.0	1.4	0.9	2.5	2.5
Cathepsins	1.3	0.0	1.15	0.0	0.0	0.5	0.0	0.0
Cystatins	0.9	0.0	0.57	0.4	0.7	1.4	0.0	0.0
5′-nucleotidase family	0.4	1.9	2.30	1.9	4.3	2.7	0.5	1.0
Prokineticins	0.4	1.0	0.00	0.0	0.0	0.0	0.0	0.0
Waprins	0.0	0.0	0.57	0.4	0.0	0.0	0.0	0.0
Phospholipases B	0.0	0.0	0.00	0.8	2.2	1.4	0.5	0.5
Venom complement C3-likes	0.0	0.0	2.87	2.3	0.0	0.0	0.0	0.0
Others								
Cellular components	42.4	32.4	22.99	8.0	10.8	7.7	15.3	6.6
Protein family not assigned	4.0	4.8	9.20	2.3	1.4	0.9	5.9	0.0

**Table 2 toxins-15-00357-t002:** Sequencing of the major toxins for each venom and their coverage according to the applied protocol.

Major Toxins Coverage	VENOMS							
	DpV		NnV		BaV		EoV	
	Alpha-Elapitoxin-Dpp2d (C0HJD7)		Cytotoxin 10 (P86541)		Bitiscetin-3 Subunit Alpha (A0A5A4WNG2)		Group III Snake Venom Metalloproteinase (E9KJY6)	
	TRYP	MELD	TRYP	MELD	TRYP	MELD	TRYP	MELD
Protein length	72 amino acids		60 amino acids		156 amino acids		515 amino acids	
Peptide coverage	59/72 = 82%	64/72 = 89%	60/60 = 100%	60/60 = 100%	104/156 = 67%	117/156 = 75%	290/515 = 56%	372/515 = 72%
#peptides	11	23	19	92	27	76	98	320
#*denovo* tags	784	1128	2596	3067	892	748	843	988
−10lgP	193.23	218.36	264.71	305.99	308.08	242.11	409.2	418.29
#spectras	125	77	387	886	645	403	442	1086
#unique peptides	7	13	1	23	23	63	7	20

## Data Availability

The data presented in this study are available in this article and Appendix A.

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
