# Peer review of "Next-Generation Sequencing for Venomics: Application of Multi-Enzymatic Limited Digestion for Inventorying the Snake Venom Arsenal"

_toxins, 2023, doi:10.3390/toxins15060357_

Round 1
Reviewer 1 Report
The manuscript has demonstrated the advantages and usefulness of multienzymatic Limited Digestion (MELD) method in comparison with classical single enzyme (Trypsin alone) method. It has a potential as a milestone change in proteomic analysis research if it is widely adopted from the society. For this, I think the manuscript might become better if the authors can appropriately address the following issues and points.
Major
1. The Venn diagram in Figure 3 displays the number of peptides and proteins identified from each venom samples, and some of the molecules are shared between each digestion protocol. The numbers, however, are literally only numbers and felt like a virtual by itself. Therefore, it is highly recommended to present the raw data of de novo peptide analyses for all of them or at least one venom sample as supplementary data.
2. In this manuscript, there is a focus on multi-enzyme digestion and de novo sequencing. During the analysis, were there any sequences of venom proteins discovered?
3. The legends in Figure 2 are not sufficiently clear and might be phrased in a little more detail. In Y-axis names, the meanings of protein groups (C) or Top Proteins (D) are not clearly addressed in the legend.
Minor
4. In Materials and methods, the section names of 4.3.1 and 4.3.2 need to be changed into italic format.
5. References need be edited according to the guideline of journal format.
Author Response
REVIEWER #1
Reviewer: Quality of English Language. I am not qualified to assess the quality of English in this paper.
Authors answer: This manuscript was revised by two English native speakers.
Reviewer: The manuscript has demonstrated the advantages and usefulness of multienzymatic Limited Digestion (MELD) method in comparison with classical single enzyme (Trypsin alone) method. It has a potential as a milestone change in proteomic analysis research if it is widely adopted from the society. For this, I think the manuscript might become better if the authors can appropriately address the following issues and points.
Authors answer: We thank the reviewer for the suggestions, as they greatly improved the quality of the manuscript.
Reviewer: Major - 1. The Venn diagram in Figure 3 displays the number of peptides and proteins identified from each venom samples, and some of the molecules are shared between each digestion protocol. The numbers, however, are literally only numbers and felt like a virtual by itself. Therefore, it is highly recommended to present the raw data of de novo peptide analyses for all of them or at least one venom sample as supplementary data.
Authors answer: We thank for this suggestion, indeed a supplementary material with this data is extremely important. We added the request material in our manuscript, therefore we prepared 4 excel files (one for each snake specie) in which we can access the peptides and proteins identified for each snake for these comparative analyses. These data were used to build the figure 3.
Reviewer: 2. In this manuscript, there is a focus on multi-enzyme digestion and de novo sequencing. During the analysis, were there any sequences of venom proteins discovered?
Authors answer: The software applied has an algorithm that allows the mutation prediction and therefore this can be useful for the identification of potential isoforms as well new toxins. Since in this manuscript we focused on a proof of concept for MELD methodology applied to venoms, we preferred to not include these analyses in this now version. However, we expect to use this information concerning the identification of new molecules combined to functional analysis in a different manuscript in the future. Some analysis is already in course, but we still need to do several evaluations to finish an idea for publication.
Reviewer: 3. The legends in Figure 2 are not sufficiently clear and might be phrased in a little more detail. In Y-axis names, the meanings of protein groups (C) or Top Proteins (D) are not clearly addressed in the legend.
Authors answer: We thank to the reviewer for this observation, and we reformulated the legend to make it clearly.
Reviewer: Minor - 4. In Materials and methods, the section names of 4.3.1 and 4.3.2 need to be changed into italic format.
Authors answer: In fact, according to the template file and guidelines from Toxins, the subsubsections should be in written in the regular format.
Reviewer: 5. References need be edited according to the guideline of journal format.
Authors answer: We revised the references and organized them according to the guideline of Toxins.
Reviewer 2 Report
In the present scientific article authors presented an alternative digestion approach to reveal the proteomics of multiple snake venoms. The new method provides more efficient to identify a great number f proteins, which is of relevance for further functional studies.
Author Response
REVIEWER #2
Reviewer: In the present scientific article authors presented an alternative digestion approach to reveal the proteomics of multiple snake venoms. The new method provides more efficient to identify a great number f proteins, which is of relevance for further functional studies.
Authors answer: We thank the reviewer for recognizing the merits of this manuscript.
Reviewer 3 Report
The study presents an innovative proteomic methodology aimed at deeply characterizing complex protein mixtures, specifically applying the MELD strategy to venomics for the first time. The results demonstrate a significant improvement in the number of sequenced (de novo) peptides, identified peptides from protein databases, and the unambiguous identification of a greater number of toxins and proteins. While I have no major concerns, I believe the manuscript can be enhanced in the following areas:
In Figure 2, the number of sequences and matches of the database search are presented for both the TRYP and MELD approaches. It would be beneficial for the authors to also include the percentage of identified proteins for both methods that have corresponding database matches. This will provide a clearer picture of the comparative efficiency of the two methods.
Figure 3 reveals that only a small proportion of peptide and protein sequences are identified using both approaches. Interestingly, for D. polylepis, the TRYP approach detected more sequences, while MELD performed better for other species. The authors should consider adding some discussion or explanations regarding this discrepancy. Potential factors or limitations that might contribute to these differences should be addressed, as well as possible implications for future research in this field.
By addressing these points, the manuscript will be more comprehensive and informative, providing a clearer understanding of the advantages and limitations of the MELD strategy in comparison to the TRYP approach.
Overall, the paper is well-written, with the authors demonstrating a strong command of the English language. The manuscript is clear, concise, and easy to follow, effectively conveying the key concepts and findings.
Author Response
REVIEWER #3
Reviewer: The study presents an innovative proteomic methodology aimed at deeply characterizing complex protein mixtures, specifically applying the MELD strategy to venomics for the first time. The results demonstrate a significant improvement in the number of sequenced (de novo) peptides, identified peptides from protein databases, and the unambiguous identification of a greater number of toxins and proteins. While I have no major concerns, I believe the manuscript can be enhanced in the following areas.
Authors answer: We thank the reviewer for recognizing the merits of this manuscript and for the suggestions, as they greatly improved the quality of the manuscript.
Reviewer: In Figure 2, the number of sequences and matches of the database search are presented for both the TRYP and MELD approaches. It would be beneficial for the authors to also include the percentage of identified proteins for both methods that have corresponding database matches. This will provide a clearer picture of the comparative efficiency of the two methods.
Authors answer: This information is already show in the Figure 4, in which a relative proportion of identified proteins are categorized according to the major toxins, minor toxins, cellular components and protein family not assigned for each snake.
Reviewer: Figure 3 reveals that only a small proportion of peptide and protein sequences are identified using both approaches. Interestingly, for D. polylepis, the TRYP approach detected more sequences, while MELD performed better for other species. The authors should consider adding some discussion or explanations regarding this discrepancy. Potential factors or limitations that might contribute to these differences should be addressed, as well as possible implications for future research in this field. By addressing these points, the manuscript will be more comprehensive and informative, providing a clearer understanding of the advantages and limitations of the MELD strategy in comparison to the TRYP approach.
Authors answer: We thank to the reviewer for this suggestion, and we reformulated the discussion for this part of our result to make it clearly.
Reviewer 4 Report
In this manuscript, the authors applied a multienzymatic approach called "MELD" to study snake venom composition, which is different from the traditional monoenzymatic-based method for sample preparation. They compare the MELD approach with the traditional method, and propose that the new method is better at identifying toxins and proteins in the snake venom.
Their work is interesting in terms of its potential to expand our knowledge about venom composition. However, some interpretations of their results should be explained more clearly to support their conclusions. Below are comments and suggestions.
1. Please provide reference for the description in line 96-97, which states that Kunitz-type toxins are rich in basic amino acid residues.
2. The text in line 118-133 is more like providing background information about experiment method, rather than discussing the results. Consider moving it to the Introduction, or stating clearly about its relation to the results.
3. In "2.2 Database search", authors explain that because DpV venom is dominated by proteins with mass range of 6-14 kDa, the venom is not beneficially affected by MELD method. However, according to Fig. 2A, the NnV venom also contains many proteins of 6-14 kDa, but significantly more sequences were detected by using MELD method. Is it possible that proteins of 6-14 kDa in either venom can not be beneficially affected by MELD, and it is the remaining proteins in NnV venom that are more readily to be detected by MELD? Or perhaps NnV venom proteins of 6-14 kDa are more readily to be detected by MELD? Explaining this contradiction can help researchers to decide when to apply MELD approach.
4. In figure 3, authors compare the number of protein sequences detected in trypsin-only, in MELD or in both approaches. The results will be more informative if authors can tell in each venom, how many (or what proportions of) toxins could be detected by both approaches (in common), and how many could only be detected by trypsin-only or MELD approach. Furthermore, what are the overrepresented biological functions of proteins or peptides exclusively detected in each method? Above information will help readers to assess the superiority of the MELD approach in detecting toxins from snake venom.
5. In the text (line 244) bradykinin-potentiating peptides are classified as minor toxins. But they are major toxins in table 1. Please confirm this.
6. In "2.3 Venom composition", authors conclude that MELD methodology improves the diversity of identified toxins. However, by counting the number of detected toxin families listed in table 1, the difference between the two approaches is not obvious. So, for each toxin family, can authors compare exact number of detected toxin protein or peptide sequences, and show how many were detected exclusively in trypsin-only or MELD approach? For toxin families which can be subdivided into sub-families (e.g 3-finger toxins), it is also interesting to know which sub-family can be detected by which approach.
7. The sentence in line 343-345 is a little confusing. Did authors want to say "isoforms for the same toxin cannot be uncovered by the monoenzymatic digestions in venomics" ? If yes, please explain why.
8. Advice on format is as follows:
"Fig" in line 157, 184 and 211 should be "Fig.";
"Figure 3." in line 207 should be "Figure 3" and not in bold type;
"C-lectin type" in line 255 should be "C-type lectin";
The format of references is inconsistent. Some journal names are abbreviated, but some are not. Title case of the journal name is inconsistent as well.
Author Response
REVIEWER #4
Reviewer: In this manuscript, the authors applied a multienzymatic approach called "MELD" to study snake venom composition, which is different from the traditional monoenzymatic-based method for sample preparation. They compare the MELD approach with the traditional method, and propose that the new method is better at identifying toxins and proteins in the snake venom. Their work is interesting in terms of its potential to expand our knowledge about venom composition. However, some interpretations of their results should be explained more clearly to support their conclusions. Below are comments and suggestions.
Authors answer: We thank the reviewer for recognizing the merits of this manuscript and for the suggestions, as they greatly improved the quality of the manuscript. We revised our manuscript and explored more the results in order to make clearly for our conclusion.
Reviewer: 1. Please provide reference for the description in line 96-97, which states that Kunitz-type toxins are rich in basic amino acid residues.
Authors answer: Reference added as requested.
Reviewer: 2. The text in line 118-133 is more like providing background information about experiment method, rather than discussing the results. Consider moving it to the Introduction, or stating clearly about its relation to the results.
Authors answer: Since we are discussing our results it is important to point out the raisons why we selected that mass spectrometer and software for our analysis, specially because we explain how the sequencing was performed.
Reviewer: 3. In "2.2 Database search", authors explain that because DpV venom is dominated by proteins with mass range of 6-14 kDa, the venom is not beneficially affected by MELD method. However, according to Fig. 2A, the NnV venom also contains many proteins of 6-14 kDa, but significantly more sequences were detected by using MELD method. Is it possible that proteins of 6-14 kDa in either venom can not be beneficially affected by MELD, and it is the remaining proteins in NnV venom that are more readily to be detected by MELD? Or perhaps NnV venom proteins of 6-14 kDa are more readily to be detected by MELD? Explaining this contradiction can help researchers to decide when to apply MELD approach.
Authors answer: Indeed, this is an important observation. Besides Naja naja snake belongs to Elapidae family, the same of Dendroaspis polylepis, which is known to be composed majority for 6-14 kDa they showed to present different outcomes depending of the applied protocol. In the Figure 2A it is possible to see that NnV present a much richer venom concerning the 28-98 KDa components compare to DpV. These can explain why NnV presented a better sequence identification and coverage even being an Elapidae snake. We added this observation in the paper. We thank for pointing out this fact for us.
Reviewer: 4. In figure 3, authors compare the number of protein sequences detected in trypsin-only, in MELD or in both approaches. The results will be more informative if authors can tell in each venom, how many (or what proportions of) toxins could be detected by both approaches (in common), and how many could only be detected by trypsin-only or MELD approach. Furthermore, what are the overrepresented biological functions of proteins or peptides exclusively detected in each method? Above information will help readers to assess the superiority of the MELD approach in detecting toxins from snake venom.
Authors answer: We thank the reviewer for their suggestion. To provide the requested information, we have created a separate file for each snake species that contains a complete comparative list of peptides and proteins, which can be accessed in the supplementary material (S2). This will enable readers to not only identify the major toxins, minor toxins, cellular components, or unassigned protein families, along with their respective Uniprot IDs, but also the peptides that were found in each digestion protocol.
Reviewer: 5. In the text (line 244) bradykinin-potentiating peptides are classified as minor toxins. But they are major toxins in table 1. Please confirm this.
Authors answer: Bradykinin-potentiating peptides was considered as a major toxin indeed, therefore we corrected this information in the text.
Reviewer: 6. In "2.3 Venom composition", authors conclude that MELD methodology improves the diversity of identified toxins. However, by counting the number of detected toxin families listed in table 1, the difference between the two approaches is not obvious. So, for each toxin family, can authors compare exact number of detected toxin protein or peptide sequences, and show how many were detected exclusively in trypsin-only or MELD approach? For toxin families which can be subdivided into sub-families (e.g 3-finger toxins), it is also interesting to know which sub-family can be detected by which approach.
Authors answer: When we refer to the 'diversity' of toxins, we are specifically referring to the classes of toxins that were identified using the MELD protocol. The differences are quite significant, with bradykinin-potentiating peptides, cathepsins, vascular endothelial growth factors, and phospholipases B being detected in some venoms only when the MELD protocol was applied. However, it is important to note that this analysis does not allow for subdivision into sub-families, as we do not have full coverage of the sequences for all toxins.
Reviewer: 7. The sentence in line 343-345 is a little confusing. Did authors want to say "isoforms for the same toxin cannot be uncovered by the monoenzymatic digestions in venomics" ? If yes, please explain why.
Authors answer: We intended to say that monoenzymatic digestions in venomics can uncover the presence of multiple isoforms of the same toxin resulting from the evolutionary process, because with the use of just one enzyme to digest we have an overlapping of fragments with the same cleavage site (e.g. -R and -K for trypsin-only digestions). We thank for the observation, and we rephrased the sentences in order to make it clearly.
Reviewer: 8. Advice on format is as follows: "Fig" in line 157, 184 and 211 should be "Fig."; "Figure 3." in line 207 should be "Figure 3" and not in bold type; "C-lectin type" in line 255 should be "C-type lectin"; The format of references is inconsistent. Some journal names are abbreviated, but some are not. Title case of the journal name is inconsistent as well.
Authors answer: We revised the whole manuscript and corrected the typos and formatted the references according to the guidelines from Toxins.
Round 2
Reviewer 1 Report
The manuscript has been well prepared and the authors have appropriately addressed to the reviewer's comments in the revised one.